# The Impact of Physical Motion Cues on Driver Braking Performance: A Clinical Study Using Driving Simulator and Eye Tracker

**DOI:** 10.3390/s23010042

**Published:** 2022-12-21

**Authors:** Sara El Hamdani, Petr Bouchner, Tereza Kunclova, David Lehet

**Affiliations:** Department of Vehicle Technology, Faculty of Transportation Sciences, Czech Technical University in Prague, Konviktská 20, 110 00 Prague, Czech Republic

**Keywords:** driving simulator, motion platform, driver behavior, performance evaluation, motion cues impact, eye tracker

## Abstract

Driving simulators are increasingly being incorporated by driving schools into a training process for a variety of vehicles. The motion platform is a major component integrated into simulators to enhance the sense of presence and fidelity of the driving simulator. However, less effort has been devoted to assessing the motion cues feedback on trainee performance in simulators. To address this gap, we thoroughly study the impact of motion cues on braking at a target point as an elementary behavior that reflects the overall driver’s performance. In this paper, we use an eye-tracking device to evaluate driver behavior in addition to evaluating data from a driving simulator and considering participants’ feedback. Furthermore, we compare the effect of different motion levels (“No motion”, “Mild motion”, and “Full motion”) in two road scenarios: with and without the pre-braking warning signs with the speed feedback given by the speedometer. The results showed that a full level of motion cues had a positive effect on braking smoothness and gaze fixation on the track. In particular, the presence of full motion cues helped the participants to gradually decelerate from 5 to 0 ms^−1^ in the last 240 m before the stop line in both scenarios, without and with warning signs, compared to the hardest braking from 25 to 0 ms^−1^ produced under the no motion cues conditions. Moreover, the results showed that a combination of the mild motion conditions and warning signs led to an underestimation of the actual speed and a greater fixation of the gaze on the speedometer. Questionnaire data revealed that 95% of the participants did not suffer from motion sickness symptoms, yet participants’ preferences did not indicate that they were aware of the impact of simulator conditions on their driving behavior.

## 1. Introduction

A driving simulator is a device with a highly sophisticated application of computer-aided kinematic and dynamic simulations that places the driver in an artificial environment intended to be a substitute for the main aspects of real driving [1]. This technology has the capacity to manipulate and control situations that are not feasible or dangerous in real traffic conditions [2], and it has prevalent use in a variety of applications, for instance, as a testbed for designing in-vehicle interfaces [3], validating autonomous vehicles [4,5], or studying drivers’ behavior [6].

Nowadays, driving simulators are increasingly being integrated by driving schools into various training processes involving four-wheels [7], trailers and rail vehicles [8], such as subways, tramways, and trains [9]. Nonetheless, research studies revealed that doubts exist regarding the objective validity of such tools in the context of transferring training to real driving [10,11,12,13]. To enhance simulator fidelity, efforts have been made in this research field to provide a more realistic sense of presence [14]. Thus, advanced artificial intelligence (AI) techniques, including deep neural network (DNN) [15], fuzzy logic [16,17,18], or genetic algorithm [19], have been exploited to optimize platform motion cueing in a high degrees-of-freedom (DOF) in the roll, pitch, and yaw axis. However, other studies underlined the high-cost issue of such developed software and hardware of motion platforms and intended to reduce simulators’ cost by decreasing the freedom to 3-DOF [20], 2-DOF [21], or even to completely static simulators [22,23,24,25,26]. This disparity between research concerning “fidelity” and “cost” is primarily due to a lack of understanding of the role of motion cues in training efficacy in the simulator. To fill this gap, it is first necessary to address the raised question of when, why, and how exactly physical motion cues could influence the driving performance of a trainee by a thorough evaluation of driver behavior.

Mclane et al. [27] conducted one of the earliest studies on motion cues and evaluated the effect of different combinations of yaw, pitch, and roll vibrations on lateral and velocity deviation in a primitive simulator while participants performed a mix of tasks involving changing velocity, changing lane, and decelerating. This study concluded that the presence of at least two motion cues could enhance driver performance, while the effect of omitting one of the three cues (i.e., roll) has an unremarkable effect. Focusing on the impact of the motion platform itself compared to a static simulator, Siegler et al. [28] evaluated driving behavior based on lateral acceleration, distance to the roadside, and linear velocity and revealed that turning the platform on prevented participants from performing overly unrealistic deceleration compared to switching the platform off. Nevertheless, the work of Colombet et al. [29] disproved the idea that classical or adaptive motion cues affected the ability of participants to perform a speed change task in a car following scenario compared to the static platform mode, but the experiments involved only three subjects, which can be considered as an invalid number of participants. In the same context, Denjean et al. [30] asked drivers to accelerate to a target speed with a hidden speedometer and the results highlighted that participants tended to underestimate their driving speed in the simulator, especially in daylight, when the acoustic feedback from the engine helped them to better perceive a vehicle motion and maintain a constant speed. Admitting that motion cues were not a concern in this study, it draws attention to the importance of gaze behavior assessment as a part of driving performance which was not included in previous motion cues related work. Moreover, another aspect that a motion platform may induce and thus affect driving performance is an increased risk of motion sickness, as stated in the work of Reinhard et al. [31], yet it was not considered in previous studies of motion cues.

Gaze estimation is a commonly used methodology [32] to obtain a better understanding of human cognition and behavior in various research fields [33], including driver’s distraction [34], safety [35], advanced driver assistance systems (ADAS) assessment [36], and development [37]. However, few papers have studied the distractions of overt visual attention of drivers in a simulator. Gomolka et al. [38] used Tobii Glasses to assess the visual attention of two distinct pilot groups with varying levels of flight training while performing flight tasks in the FNTP II MCC simulator. In the driving simulator context, Le et al. [39] used an eye tracker to observe and compare drivers’ eye movements in two different situations: in a simulator and on a real road and captured a higher level of cognitive distraction in the naturalistic case compared to the simulator. Nonetheless, eye sensors have not been employed in the context of studying the impact of simulator conditions, such as motion cues, on the trainee’s cognitive distraction.

All the surveyed papers had clear limitations, used a primitive simulator, included an insufficient number of participants (less than 10), did not consider the motion sickness effect, or used a complicated list of driving tasks which do not help to provide accurate explanations of how much and in which way motion cues impact driving performance. To the best of our knowledge, no previous work has presented a comprehensive study of motion cues’ effect on driving behavior that evaluates their importance during training in a simulator. To address this gap, we thoroughly study the impact of motion cues on braking at a target point as an elementary behavior [40] that reflects the overall drivers’ performance. The main purpose of our study is to evaluate if the feedback given by motion cues has a positive impact on the driver’s speed perception and hence on braking performance, gaze focus, and driving comfort compared to the insight provided by the speedometer and warning signs. The novelty of the present paper is the use of an eye-tracking device to assess gaze behavior in addition to evaluating driving performance in the simulator and considering participants’ feedback. Moreover, we compare the effect of different motion levels first without and then with the pre-braking warning signs.

## 2. Materials and Methods

### 2.1. Participants

A total of 24 drivers in the age range of 20–65 years (*M* = 29.83 years, *SD* = 11.30 years), 5 females and 19 males, participated in the experiment. The participants included both employees and students of CTU in Prague, as well as volunteers from non-academic backgrounds; all recruited were recruited via the university’s email service. Possession of a valid driving license of at least category B in Europe and an age between 19–70 years were required to participate in the study. The driving experience of the participants ranged from 0.5 to 50 years (*M* = 11.42 years, *SD* = 11.90 years), with 58.33% of the participants driving at least 4 times a week and the rest a few times a month. Further, 54.17% of the participants had previous simulator experience. One participant was unable to complete the experiment due to motion sickness; hence, the data of 23 participants were processed when evaluating the simulator data.

### 2.2. Instruments

#### 2.2.1. Simulator and Motion Platform

The experiment was conducted in the laboratory simulator shown in Figure 1. The main hardware part consists of the front part of the Škoda Superb III with automatic transmission [41]. The car’s dynamics model is equivalent to a front-wheel drive European middle-class car and it was developed in our laboratory (R&D 4.0 LAB) at the Technical University in Prague [42]. The vehicle is placed on a motion platform that is optimized with respect to the driving scenario and copies road irregularities and curves as well as the acceleration with high accuracy. Three full HD TVs positioned to cover 100% of the visible area from the front and both side windows of the vehicle were used to project the scenario. These TVs are rigidly connected to the moving structure and thus replicate the movement of the entire driving simulator. The software part of the simulator consists of a virtual reality (VR) engine responsible for generating 3D graphics and spatial audio for the physical engine.

The motion platform, as described in Figure 2a, consists of six electric motors and six actuators mounted between the upper frame and the bottom frame of about two meters (Figure 2a). The combination of this hardware constitutes a platform able to move in the pitch, roll, and yaw axis, as illustrated in Figure 2c, enabling the simulator motion cues in six degrees of freedom 6-DOF. Specifically, it is a hydraulic platform supplied by Pragolet, s.r.o. [43] and its movements are controlled by an optimal motion cues algorithm consisting of the washout filter, the vehicle’s mathematical–physical model, and the kinematic transformation of the actuators’ position, as previously described by our colleagues in [44].

#### 2.2.2. Eye Tracker

To observe visual behavior in our experiment, we used the Tobii 2 50 Hz wireless eye tracker equipped with four eye tracking cameras monitoring the participants’ eyes and one FullHD camera monitoring the environment in front of the participants, as illustrated in Figure 3 [45]. Tobii is a binocular eye tracker based on a video-oculography method that enables video detection of pupil position based on the reflection of infrared light shining on the surface of the eye. Accordingly, the proband’s gaze direction vector is calculated and mapped onto the image captured by the eye tracker’s front camera with a vector accuracy of approximately 0.6.

The Tobii Pro Lab application based on the velocity-thres hold identification (I-VT) fixation classification algorithm is used to process the raw data recorded by the eye tracker and classifies the individual states of the eye into saccade, fixation, and blink based on the velocity of the sampled eye movement compared to a threshold parameter. In this context, eye movements below the velocity threshold are classified as fixations, while eye movements with higher velocities are considered saccades. In our laboratory, the velocity threshold for all experiments is generally set to 30°/s, a value frequently used in research [46]. 

#### 2.2.3. Questionnaire

The participants’ feedback was obtained via a structured questionnaire, and then their subjective opinions were evaluated. After each run of the experiment with a different combination of road scenario and motion cues level, we asked the participants to rate their comfort with motion level and the difficulty of the task on a five-point Likert scale. Moreover, the participants were asked to select the area that caught their attention the most during the run, either the speedometer, the warning signs, or the track lane. After completing three runs of the experiment, when the participants had tried all levels of the motion platform, they were asked how they perceived the effect of the motion cues and how they thought it helped them in the task. To verify if the driving performance was affected by any motion sickness, the participants selected the statement describing their condition after the experiment based on the MIsery SCale (MISC) [47] presented in Table 1.

### 2.3. Data Analysis

Data from the driving simulator, eye tracker, and questionnaire were used as input for analysis, applying both descriptive and inferential statistics to obtain outputs and meet the objective of the present study. Data from the driving simulator were processed, filtered from the motion-sick participant, and the mean speed profiles over distance were generated. The Tobii Pro Lab application was used to analyze the eye tracker data. The data were manually checked, including the AOI settings, and the proportion of total time in AOIs was obtained. Data from the questionnaire were used for general information about the participants, a descriptive overview of the participants’ opinions, and for statistical evaluation. Based on the characteristics of the data (group comparison, meeting normal distribution, homogeneity of variances, etc.), a parametric one-way analysis of variance (ANOVA) test was performed on the data regarding participants’ preferences.

### 2.4. Experimental Design

The primary concern of the present research is to study the effect of motion cues on driving behavior with a focus on the ability to accomplish smooth braking to a target point under different motion cues levels compared to static simulator driving. Accordingly, the braking behavior can reflect drivers’ ability to correctly estimate their current speed in the simulator and interpret the distance to a fixed point on the road displayed on 2D screens.

For our experiment, a driving scenario consisting of a two-lane, two-way rural road of 4200 m in length was designed, as illustrated in Figure 4a. In this scene, the participants performed a braking task at a target point from a driving speed of 120 km/h. As shown in Figure 4b, the target stopping point involves a railway crossing on the road, and the participants were instructed to stop the vehicle directly at the stop line located in front of the closed crossing barrier marked with a STOP sign and flashing lights.

In order to test the braking behavior under different road conditions, a second scenario was created with one difference—by placing warning signs, as shown in Figure 4c, to alert the drivers of an upcoming railway crossing and prepare them to start decreasing their speed. As shown in Figure 4d, three different “Countdown Markers” with stripes were placed at distances of 80 m (three stripes sign), 160 m (two stripes sign), and 240 m (one stripe sign) in front of the railway crossing.

The warning sign distances used towards the end of the scene provide the time required to gradually decelerate to the target from a high initial speed (120 km/h) according to the AASHTO [48] stopping sight distance Equation (1):(1)$$S=\frac{\left(0.278\times t\times v\right)+v²\;}{254\times \left(f+G\right)}$$
where “*S*” is stopping distance in (m), “*t*” is perception–reaction time in seconds with 2.5 s for the worst-case scenario, “*v*” is speed of the vehicle in km/h, “*G*” is grade or slope of the road expressed as a decimal, and “*f*” is the coefficient of friction between the wheels and the road: 0.7 for a dry road and from 0.3 to 0.4 for a wet road.

### 2.5. Experiment Procedure

The experiment was carried out in a darkened and quiet room. Each participant was asked for general information at the beginning of the experiment, including age, driving experience, etc. Afterward, all the participants were given the necessary time to adapt to driving in the simulator, during which they tested all the relevant vehicle functions using a trial scenario, as recommended by previous research [49,50]. The inexperienced participants needed 5 to 10 min to adapt, while the experienced participants needed approximately 1 to 3 min. After the test drive, the participants were asked to put on the eye-tracking glasses, and the device was calibrated.

As aforementioned, the participants were asked to drive straight down the road (Figure 4a), accelerate to 120 km/h, and stop at the stop line in front of the railway crossing (Figure 4b). Each participant repeated the same task six times in each run under different road and motion conditions. As summarized in Figure 5, these six conditions were based on a combination of two traffic scenarios (with and without warning signs) and three different motion levels, namely, no motion cues, low level of motion, and high level of motion.

Before each round, the motion platform’s parameters were tuned to an appropriate level of motion according to the measurement plan. The platform’s motion is controlled by the scale function in Figure 6, defined by the maximum acceleration and gain (factor) in three axes and the maximum angular velocity in the *x* and *y* axes. As summarized in Table 2, the parameters were set to 0 for the static level (gain is not considered in this case). For the high level of motion, the parameters were set to the higher value recommended for this simulator with minimal motion sickness effect. These values were then halved for the low level of motion. Each participant’s measurement took an average of 40 min, including adaptation to the simulator, device calibration, six rounds of the driving task (2.5 min each), changing the motion parameters, and completing the questionnaire described in the following subsection.

## 3. Experimental Results

In this section, the results of six runs of the experiment (under different combinations of road and motion conditions) are presented, obtained from three different types of data: simulator data, eye tracker data, and questionnaire data.

### 3.1. Simulator Data

The participants’ speed behavior in the simulator was analyzed under six combinations of road and motion conditions. The mean speed profiles at different distances to the final stop line are plotted in Figure 7.

As shown in Figure 7a, the six mean speed profiles varied from the start point (4200 m) to the end point (0 m) following the same general pattern corresponding to the task design, which consisted of accelerating to ≅120 km/h (>33, 33 ms^−1^), maintaining speed and stopping at the stop line.

Nevertheless, Figure 7b zooms in on the last 1000 m before the stop line, during which the different profiles started slowing down and preparing for full braking and shows the difference between the adopted speed behaviors under different conditions. The graph shows that both “No motion”, either with signs or without signs, as well as the “Mild motion without signs” profiles, tended to gradually and significantly reduce speed, while the three other profiles maintained a high speed, leading to low braking performance. Overall, the “Full motion” profiles showed the smoothest change in speed between 1000 m to 240 m to the stop line. “Full motion with signs” achieved slightly better performance, which refers to the positive impact of the feedback given by both: the high level of motion cues and warning signs on the driver’s speed perception.

To further compare different braking behaviors, we focused on the last 240 m before the stop line (as presented in Figure 7c), considered as the actual braking distance based on Equation (1), and where the warning signs were placed. As expected, both “Full motion” profiles continued to decelerate gradually and achieved smooth braking, but with equivalent performance in this area with no impact of warning signs. This means that the positive effect of high motion cues on speed behavior outperforms the feedback provided by warning signs. Similarly, the “Mild motion” profiles also showed a smooth change in speed but of higher values compared to “Full motion”.

Unexpectedly, we noticed that the “Mild motion with signs” profile was less smooth (changing speed from 22 to 0 ms^−1^) than the “Mild motion without signs” profile (changing speed from 10 to 0 ms^−1^), indicating that warning signs had a negative impact on speed behavior under the mild motion conditions. The worst speed performance was observed for both “No motion” profiles. In fact, the “No motion with signs” profile produced the worst change of speed (from >25 to 0 ms^−1^) over a short distance of 240 m, while the “No motion without signs” profile started from a lower speed (20 ms^−1^) but delayed a significant reduction in speed until the last 80 m before the final stopping point, leading to hard braking. Overall, the braking performance increased significantly with the level of motion; the warning signs had no significant impact in the presence of the high motion level and had a negative impact with the mild motion level.

### 3.2. Eye Tracker Data

The participants’ gaze behavior was evaluated at three levels of motion by analyzing the eye tracker data, separately for the “Without signs” and “With signs” scenarios.

#### 3.2.1. “Without Signs” Scenario

For the first scenario, “Without signs”, our methodology consisted of classifying the gaze interest into the “Track” area representing the focus on the road, and the “Dashboard” area representing the focus on the speedometer according to the areas of interest (AOI) map presented in Figure 8a. Therefore, we extracted gaze fixation maps of 23 participants during the driving task under “No motion” (Figure 8c), “Mild motion” (Figure 8d), and “Full motion” (Figure 8d) conditions. The three gaze maps show a slight increase in focus on the “Track” compared to the “Dashboard” area as the motion cue level increases.

To quantitatively evaluate the gaze interest, we analyzed the share of average fixation time between the “Track” and the “Dashboard” areas, as presented in Figure 8b. As anticipated in our research assumptions, the graph shows a slight and gradual increase in fixation time for the “Track” area compared to the “Dashboard” area as the motion level increases, respectively: 81.89% under “No motion”, 83.29% under “Mild motion”, and 85.36% under “Full motion” conditions. Correspondingly, the increase in the motion level negatively and gradually impacted the fixation time on the “Dashboard” area. The results indicate that the participants’ perception of speed rather relied on the motions cues feedback, especially when at a high level, compared to the feedback provided by the speedometer. Consequently, the presence of high-level motion cues may help improve safety by reducing the distraction caused by monitoring the HMI and allowing the driver to concentrate more on the track.

#### 3.2.2. “With Signs” Scenario

For the “With signs” scenario, our methodology relied on the AOI classification according to the map presented in Figure 9a consisting of the “Warning signs” area in addition to the “Dashboard” and the “Track” areas. Similarly to the first scenario, we extracted the gaze maps of the warning signs scenario under the three levels of motion cues, respectively under “No motion” (Figure 9c), “Mild motion” (Figure 9d), and “Full motion” (Figure 9d) conditions.

As expected, the graph indicates a slight gradual decrease in fixation time in the “Warning signs” area, respectively: 6.25% under “No motion”, 6.15% under “Mild motion”, and 5.01% under “Full motion” conditions, meaning that the stronger the motion level, the lower the driver’s interest in signs’ feedback; hence the best eye fixation on the “Track” (82.92%) was achieved under full motion conditions. However, the presence of warning signs had a negative impact on the focus (eye fixation) on the track under the “Mild motion” condition. In other words, the low level of motion cues is not strong enough to help the drivers correctly perceive their actual speed, and with the presence of the warning signs, the participants tended to check the actual speed on the speedometer more.

### 3.3. Questionnaire Data

We evaluated the feedback of ≈24 participants on the experiment by analyzing data from the questionnaire. As aforementioned, one of the participants was not able to complete the six rounds of the experiment, and hence we considered the answers concerning the part that was accomplished. Figure 10 presents the mean scores given by the participants for the different motion cues’ levels using a five-point Likert scale according to their approximation to the real car motion cues. Unlike the simulator data, the graph of the mean scores shows a major preference for the “Mild motion” level with a mean score of 3.57 pts with a standard deviation (SD) = 0.992. Consequently, the “Full Motion” level had a lower score of 2.65 pts with an SD = 1.26, while the “No Motion” level had the lowest score of 2.30 pts with an SD = 1.19. In other words, the participants did not prefer driving in the static simulator, although they did not favor the high level of motion cues.

To assess the independence of the participants’ preferences for the motion level (the null hypothesis H_0_ (2)), we conducted a one-way ANOVA test on one independent variable, which is the participants’ preference for “No Motion” µ_N–M_, “Mild Motion” µ_M–M_, and “Full Motion” µ_F–M_ presented in Figure 10:H_0_: µ_N–M_ = µ_M–M_ = µ_F–M_(2)

According to the results of the ANOVA test summarized in Table 3, the probability of confirming the null hypothesis (H_0_) is very low, (F = 7337) and the *p*-value is less than the threshold of 5% (*p*-value = 0.00133 < 0.05). The latter means that there was a difference between the opinions of the drivers about the levels of motion and their approximation to reality based on their subjective feeling. Consequently, the subjects’ perception and preference of the motion level were not correlated with their performance under these conditions.

The motion sickness state of the participants was checked during each run and at the end of the experiment, the drivers rated their condition based on the MISC 10-point scale [47]. Figure 11 presents the distribution of the total number of participants according to their scores of motion sickness and shows that most participants (95%) were assigned a low MISC score (≤3 pts), with only two participants having a score higher. It is worth noting that the highest score was obtained by the participant who did not complete the experiment, and therefore the simulator and eye tracker data were excluded. In addition, 90% of the participants reported that their state of comfort had improved with each run of the experiment, while the others reported that their state had not changed. Therefore, we can conclude that the output of our motion cues evaluation was not affected by motion sickness.

## 4. Discussion

During our experiment, the participants achieved smoother braking in the “Full motion” conditions compared to the “Mild motion” and the “No motion” levels, both in the presence and absence of warning signs, especially in the last 240 m before the final stop line. As expected, the high-level motion cues provide the participants with sufficient feedback about the actual driving speed and have a positive effect on driving performance in the simulator, confirming the findings of previous studies [28,29]. Moreover, the “Full motion” mode also had a slightly positive impact on the gaze behavior of the drivers in terms of focusing on the road and reduced their need to look at the speedometer or warning signs, suggesting that motion feedback is more effective on drivers’ perception compared to the speed information provided by a speedometer. These results confirm the positive impact of the driving simulator conditions on the visual concentration of the trainees, which is related to the findings of an eye-tracker-based study conducted by Le et al. [39], showing that participants had a lower level of visual distraction in the simulator compared to the real world situation on the road. However, eye sensors have not been widely exploited in studies concerning driver behavior in the simulator.

Contrarily, the braking performance of the participants at the “Mild motion” was better in the first scenario “Without warning signs” compared to the second “With warning signs”. Similarly, “Mild motion” cues had a negative impact on the participants’ gaze behavior in the presence of warning signs and increased their interest in watching the speedometer. One explanation for the low performance under these conditions is that the mild level of motion cues causes an underestimation of the actual driving speed, and with the warning signs, the participants tend to check the HMI to make sure they are driving at an adequate speed.

The platform in the static mode had the worst impact on both braking performance and the gaze behavior of the drivers, especially in the “Without warning signs” scenario. Likewise, the “No motion” level had the lowest score in terms of its approximation to reality according to the subjective assessment of the participants. Nevertheless, the “Mild motion” level was rated better by the participants than the “Full motion” level, which is not correlated with their driving performance under these conditions. Thus, we can state that the drivers have an incorrect awareness of the impact of driving conditions on their performance, and their subjective preferences do not represent a correct perception of the situation in the simulator. This statement is supported by the findings of the study of Talsma et al. [47], in which they recommended basing the evaluation on objective or physiological data of the participants.

The evaluation of the state of sickness showed that most of the participants (95%) had an extremely low level of sickness (MISC score ≤ 3 pts). In addition, 90% of the participants reported that their comfort in the simulator improved with each run. As a result, we concluded that driver behavior was not affected by motion sickness. Nonetheless, it should be noted that the eye-tracking outputs were not sufficiently compared with the findings because, to our knowledge, no previous research has evaluated the effect of motion cues on drivers’ gaze behavior. In conclusion, the results of our study can have a broad impact on research and industries related to various driving simulators by increasing the knowledge of the impact of simulator conditions and parameters on the quality of training, precisely to support the use and improvement of motion platforms with a full level of motion cues.

## 5. Conclusions

The present paper aimed to study the impact of the feedback given by the simulator platform’s motion cues on the driver’s behavior. In our experiment, we asked 24 participants to brake at a target point after reaching a certain speed under different motion levels (“No motion”, “Mild motion” and “Full motion”) and scenarios (“With warning signs” and “Without warning signs”). In addition to the participants’ subjective feedback and their motion sickness state from the questionnaire, we evaluated the participants’ driving performance in the simulator in terms of speed change behavior during braking, and the gaze behavior from the eye tracker in terms of fixation time on three areas of interest, namely “Track”, “Speedometer”, and “Warning signs”. The results showed that the participants achieved the smoothest braking with greater eye focus on the track in the “Full motion” level without any significant impact of the feedback given by the speedometer and warning signs. However, the “Mild motion” level outperformed the “No motion” level in the “Without warning signs” scenario but had a negative effect with the presence of warning signs due to underestimation of the actual speed and the need to have a proper overview of the speedometer. Questionnaire data revealed that most participants did not suffer from motion sickness symptoms, yet participants’ preferences did not indicate that they were aware of the impact of simulator conditions on their driving behavior. As none of the previous studies have used an eye tracker in a similar context, we consider further investigating the impact of motion cues on gaze behavior in more complex driving scenarios.

## Figures and Tables

**Figure 1 sensors-23-00042-f001:**
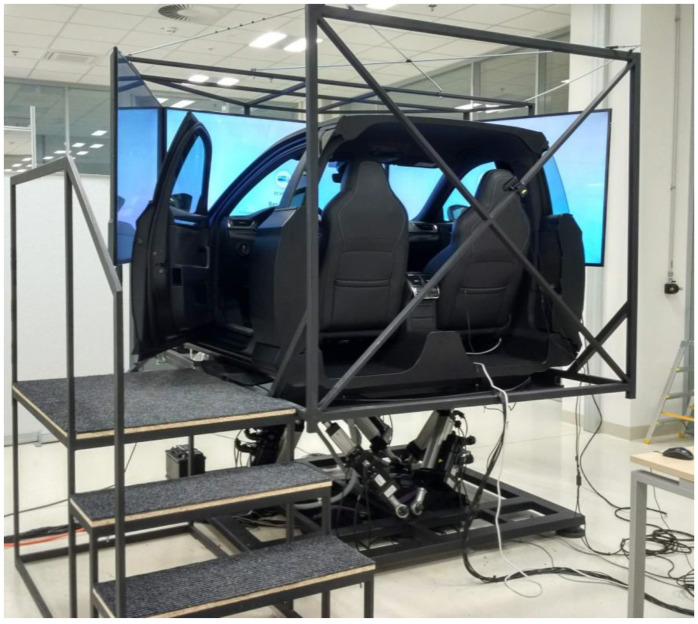
The driving simulator used for the present study developed in our laboratory (R&D 4.0 LAB) at the Faculty of Transportation Sciences, CTU in Prague.

**Figure 2 sensors-23-00042-f002:**
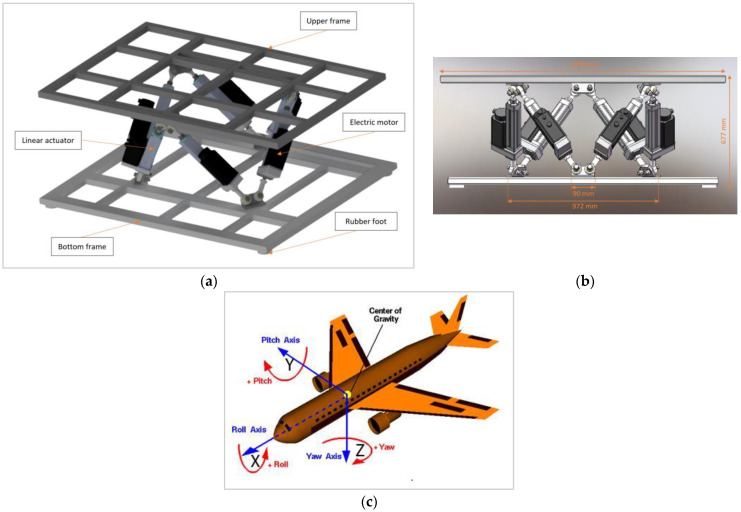
Illustration of the motion platform design with six degrees of freedom of the used simulator: (**a**) illustrates the hardware components of the platform; (**b**) illustrates the size of the platform; (**c**) illustrates the pitch, roll, yaw axes enabling 6-DOF motion of the platform.

**Figure 3 sensors-23-00042-f003:**
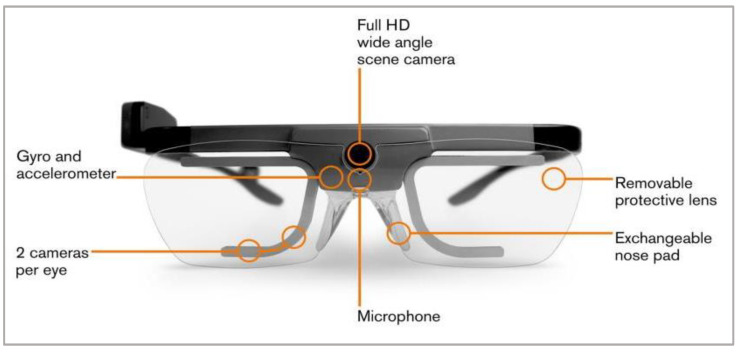
Illustration of the construction of the eye tracker used in our experiment [45].

**Figure 4 sensors-23-00042-f004:**
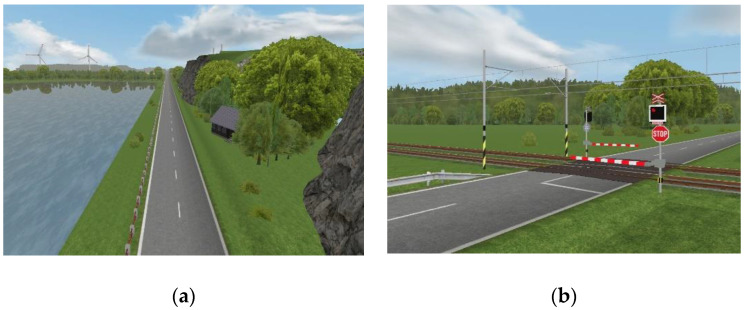
The driving scenario used for our experiment: (**a**) illustrates the track road; (**b**) illustrates the stop line representing the target point; (**c**) illustrates the warning signs used as a deceleration signal in the scenario with warning signs and (**d**) illustrates the three “Countdown Markers” placed at distances to the Stop line.

**Figure 5 sensors-23-00042-f005:**
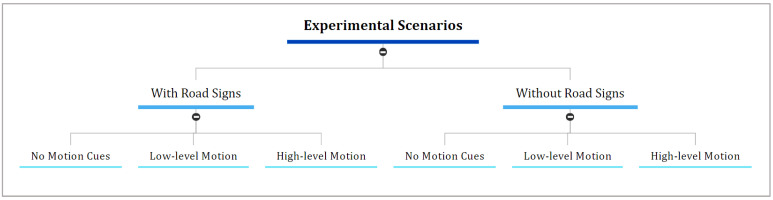
Summary of the six tasks’ conditions based on a combination of different track designs and motion cues levels used in our experiment.

**Figure 6 sensors-23-00042-f006:**
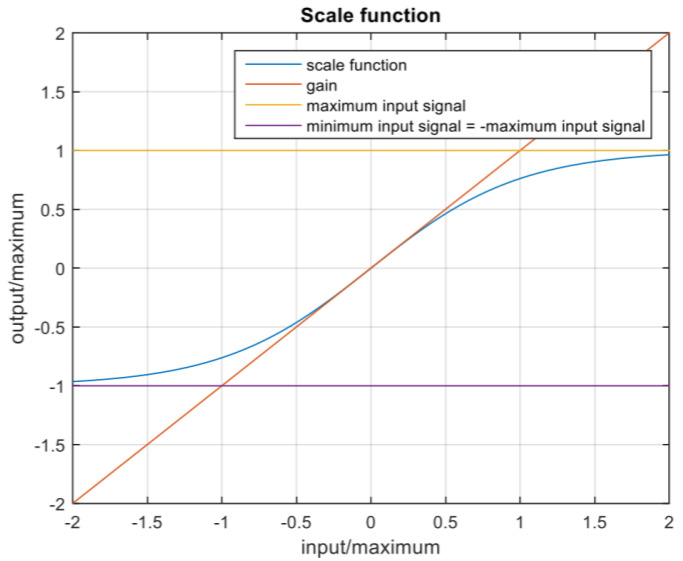
Representation of the scale function (defined by the parameters in Table 1) which scales the input signal from the mathematical model of the simulated vehicle to the motion platform.

**Figure 7 sensors-23-00042-f007:**
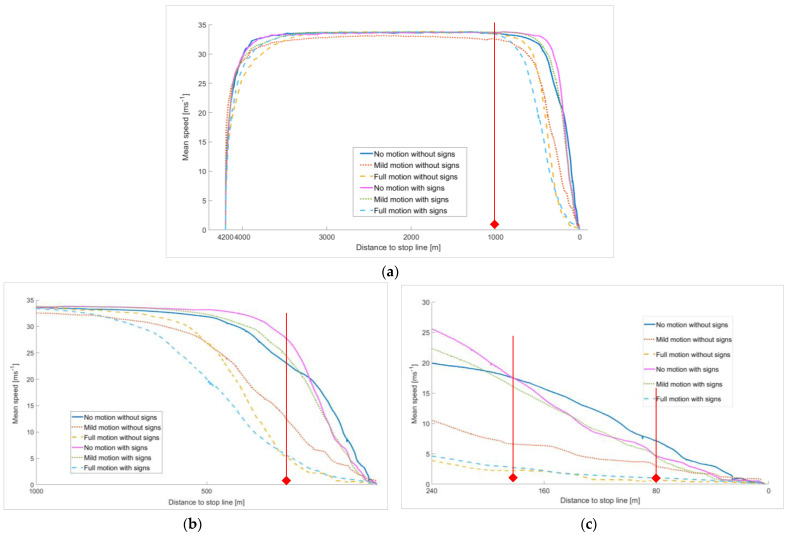
Mean speed profiles at various distances from the stop line: (**a**) 4200 m before Stop line, (**b**) 1000 m before Stop line and (**c**) 240 before Stop line. The vertical lines in red were inserted as reference points to indicate a significant change in speed behavior in the same area.

**Figure 8 sensors-23-00042-f008:**
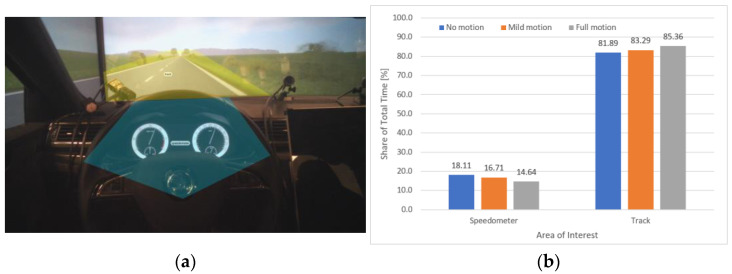
Eye tracker data analysis in the “Without signs” scenario. (**a**) The areas of interest (AOI) map consisting of the “Track” area highlighted in yellow and the “Dashboard” area in green. (**b**) The total share of time of gaze focus on the different AOIs under different motion conditions. (**c**–**e**) are the “Gaze Maps”, respectively, under “No Motion”, “Mild Motion”, and “Full Motion” conditions.

**Figure 9 sensors-23-00042-f009:**
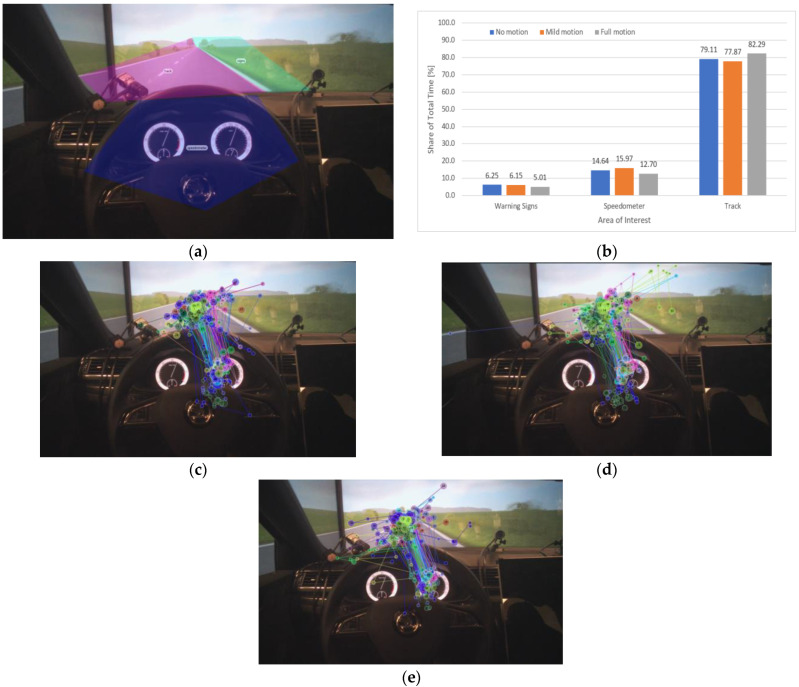
Eye tracker data analysis in the “With signs” scenario. (**a**) The area of interest (AOI) map consisting of the “Track” area highlighted in pink, the “Dashboard” area in blue and “Warning signs” angle in green. (**b**) The total share of time of gaze focus on the different AOIs under different motion conditions. (**c**–**e**) are the “Gaze Maps”, respectively, under “No Motion”, “Mild Motion”, and “Full Motion” conditions.

**Figure 10 sensors-23-00042-f010:**
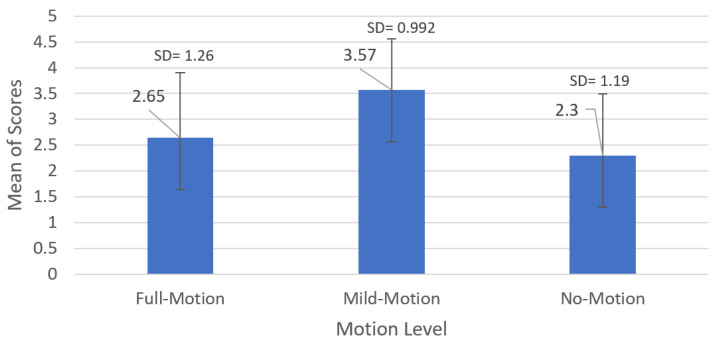
Mean scores given by the participants to the different levels of the motion platform’s cues.

**Figure 11 sensors-23-00042-f011:**
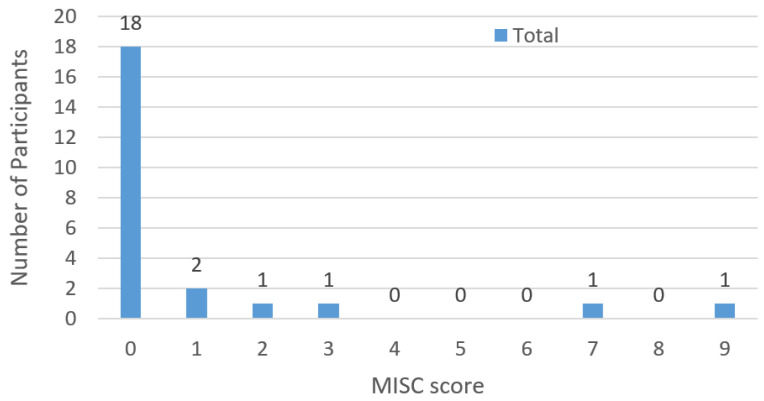
Distribution of the participants according to their motion sickness state based on MISC score.

**Table 1 sensors-23-00042-t001:** MIsery SCale (MISC) included in the questionnaire for evaluating the motion sickness state of the participants in our experiment.

Symptom		Score
No problems		0
Uneasiness (no typical symptoms)		1
Dizziness, warmth, headache, stomach awareness, sweating …	vague	2
	slight	3
	fairly	4
	severe	5
Nausea	slight	6
	fairly	7
	severe	8
	retching	9
Vomiting		10

**Table 2 sensors-23-00042-t002:** Summary of the motion platform parameters that define scale function (Figure 6) tuned for different motion conditions, namely “No motion”, “Mild motion”, and “High motion” levels of motion cues.

Motion Parameter	Description	Axis	Motion Level		
			No Motion	Mild Motion	High Motion
Maximum Acceleration [rad/s^2^]	Limit for input linear acceleration, which defines the Scale function.	*x_Acc_*	0	0.7	1.5
		*y_Acc_*	0	0.55	1.1
		*z_Acc_*	0	0.7	1.5
Maximum Angular Velocity [rad/s]	Limit for input angular velocity, which defines the Scale function.	*x_AV_*	0	0.07	0.15
		*y_AV_*	0	0.07	0.15
Gain [-]	Multiplication factor (gain) of the linear acceleration that defines the Scale function.	*x_G_*	-	0.45	0.9
		*y_G_*	-	0.1	0.2
		*z_G_*	-	0.15	0.3

**Table 3 sensors-23-00042-t003:** Summary of the results of the one-way ANOVA test conducted on the mean scores assigned to three levels of motion in Figure 10.

	Df	Sum Sq	Mean Sq	F	*p*-Value ^1^
Motion Level Score	2	19.51	9.754	7.337	0.00133
Residuals	66	87.74	1.329		

^1^*p*-value (<0.05) is statistically significant.

## Data Availability

Not applicable.

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
