# Peer review of "The Impact of Physical Motion Cues on Driver Braking Performance: A Clinical Study Using Driving Simulator and Eye Tracker"

_sensors, 2022, doi:10.3390/s23010042_

Round 1
Reviewer 1 Report
The manuscript is having the good work done, however
the Authors should work more on the literature review and collect the recent works done on driver remarkable findings and also what are the recent eye tracking systems used in the existing Systems.
The Authors may refer to some of the recent works like
Optimized feature extraction for precise sign gesture recognition using self-improved genetic algorithm
1. A Neural Network Approach for Indoor Fingerprinting-Based Localization
It is very much required if the authors can clearly indicate on the process of eye tracking and functioning of drivers iability in terms of braking Systems.
Authors should the simulators for the correct tracking system and driver braking Technology.
The Authors could consider the equation 3 and 4 and why they have used them, they should provide a justification for the same.
Author Response
The Report to Reviewer 1 is attached.

Reviewer 2 Report
1) I think the keywords are too many. Please use most important and relevant keywords in the abstract.
2) Line 82, this is “Device”, not “Divice”.
3) Line 84, this is “Moreover”, not “Moreove”. Please check carefully throughout the text.
4) Please place Figure 1 and 2 after 2.1.
5) Line 86 to 91: Why did you report the results of this study at the end of Introduction?
6) Please reconsider the arrangement of section 2 (Materials and Methods). I suggest that you begin with Participants, Simulator and Motion Platform, and then Instruments (including eye-tracker and questionnaires), ……
7) I did not observe a section regarding “Data Analysis” in Materials and Methods. How did you explained which statistical tests were used in this study?
8) I think Discussion needs to be strengthened, mostly regarding the comparison of your findings with previous findings.
9) How are the practical implications of your study. I think this study could have important practical implications. Please note them in the Discussion.
Good luck
Author Response
The responses to the Reviewer 2's comments are insered in the Report attached.

Reviewer 3 Report
Thanks for the opportunity to review this interesting article. My recommendations are included in the attached document.

Author Response
The responses to the Reviewer 3's comments are insered in the Report attached.

Round 2
Reviewer 2 Report
Thank you for your revision.